# "Non-Eloquent" brain regions predict neuropsychological outcome in tumor patients undergoing awake craniotomy

Muhammad Omar Chohan[1], Ranee Ann Flores[2], Christopher Wertz[2], Rex Eugene Jung[2]*

1 Department of Neurosurgery, University of Mississippi Medical Center, Jackson, Mississippi, United States of America, 2 Department of Neurosurgery, University of New Mexico Health Sciences Center, Albuquerque, New Mexico, United States of America

* rex.jung@gmail.com

**Data Availability Statement:** All relevant data are within the paper and its Supporting Information file.

## Abstract

Supratotal resection of primary brain tumors is being advocated especially when involving "non-eloquent" tissue. However, there is extensive neuropsychological data implicating functions critical to higher cognition in areas considered "non-eloquent" by most surgeons. The goal of the study was to determine pre-surgical brain regions that would be predictive of cognitive outcome at 4–6 months post-surgery. Cortical reconstruction and volumetric segmentation were performed with the FreeSurfer-v6.0 image analysis suite. Linear regression models were used to regress cortical volumes from both hemispheres, against the total cognitive z-score to determine the relationship between brain structure and broad cognitive functioning while controlling for age, sex, and total segmented brain volume. We identified 62 consecutive patients who underwent planned awake resections of primary (n = 55, 88%) and metastatic at the University of New Mexico Hospital between 2015 and 2019. Of those, 42 (23 males, 25 left hemispheric lesions) had complete pre and post-op neuropsychological data available and were included in this study. Overall, total neuropsychological functioning was somewhat worse (p = 0.09) at post-operative neuropsychological outcome (Mean = -.20) than at baseline (Mean = .00). Patients with radiation following resection (n = 32) performed marginally worse (p = .036). We found that several discrete brain volumes obtained pre-surgery predicted neuropsychological outcome post-resection. For the total sample, these volumes included: left fusiform, right lateral orbital frontal, right post central, and right paracentral regions. Regardless of lesion lateralization, volumes within the right frontal lobe, and specifically right orbitofrontal cortex, predicted neuropsychological difference scores. The current study highlights the gaps in our current understanding of brain eloquence. We hypothesize that the volume of tissue within the right lateral orbital frontal lobe represents important cognitive reserve capacity in patients undergoing tumor surgery. Our data also cautions the neurosurgeon when considering supratotal resections of tumors that do not extend into areas considered "non-eloquent" by current standards.

**Funding:** The author(s) received no specific funding for this work.

**Competing interests:** The authors have declared that no competing interests exist.

## 1. Introduction

Neurosurgical management of primary brain tumors emphasizes extent of resection (EOR) as the cornerstone in achieving favorable oncological endpoints. Careful surgical resection of brain tumors has evolved over the past 100 years with improvements in pre- and intra-operative brain mapping, including under awake conditions, to minimize functional cost for oncological benefit [1]. However, much of the functional mapping, especially intraoperative assessment, has been limited by tailored, time constrained tasks like motor and speech testing. While uniquely informative, these assessments do not provide a thorough evaluation of patient's global cognitive abilities, critically linked to intact brain tissue [2].

When resections are performed on what were classically considered to be "non-eloquent" areas, it is generally understood that these regions do not represent commonly mapped behaviors of sensorimotor, vision and speech [3], thus allowing for more radical resections [4, 5]. However, there is extensive neuropsychological, neuroimaging, and neuropsychiatric data implicating functions critical to higher cognition in areas previously considered to be "non-eloquent" by most surgeons. These functions include executive function, personality, motivation, attention [6], memory [7], intelligence [7, 8], and aspects of creative cognition [9–11] that are widely distributed across prefrontal, temporal and parietal regions [9, 11, 12] and their associated subcortical networks [13, 14].

Despite its impact on post-surgical rehabilitation and prognosis [15, 16], and a strong advocacy as an objective outcome measure [17, 18], relatively few studies have looked at neuropsychological effects of brain tumors [19–22]. Fewer yet have established "brain-behavioral" correlates, linking regions outside, but near, the tumor zone with functional capacity integrated through established brain networks (i.e., executive control, default, etc.) Moreover, most studies have focused on treatment related sequalae on global cognitive measures [15, 23–26] (for a review, see [27]), with visuospatial and fluency as the most frequently affected functions [21, 22, 28], at least partly associated with depression [19, 25]. While most studies report some improvement in cognitive functions after surgery [19, 22], they lack long-term follow-up [21, 22, 27–29], and still others report continued decline over time, which was interpreted to be multifactorial [15, 26].

Modern structural neuroimaging techniques have helped to establish "brain-behavioral" correlates in both health and disease, with recent studies linking cortical volume measures to cognitive functioning in: fronto-temporal and Alzheimer's dementia [30], attention deficit disorder [31], first episode psychosis [32], systemic lupus erythematosus [33], alcohol use disorder [34], and mild traumatic brain injury [35]. Correlates between cortical volume and cognition have also been made in adolescent development [36], middle age cognitive reserve [37], and healthy cognitive aging [38]. Finally, our group has worked extensively in this area of research, having previously described cortical volume correlates across a diverse number of cognitive capacities including intelligence [39, 40], response time [41], visuo-spatial learning [42], creativity [43–47], personality [43, 48], and imagination ability [49]. The convergence of these studies, across both disease and health, implicates cortical volume as an important biomarker of current cognitive functioning.

The current study was designed to establish "brain-behavioral" relationships in a cohort of tumor patients described previously [50]. We explore regions both within the tumor zone, as well as within the broader EOR, in a cohort representing a wide range of brain tumors (e.g., gliomas, cavernous malformations) who had cognitive testing both before and after resection. The goal of the study was to determine pre-surgical brain regions that would be predictive of cognitive outcome post-surgery, which could serve to guide clinicians in their determination of optimal EOR, while preserving brain regions and networks critical to cognitive outcome.

Given well-stablished "brain-behavior" relationships obtained from the neuropsychological literature [51], we hypothesized that: global neuropsychological functioning would be related to structures broadly distributed throughout the brain, including regions within the frontal, temporal, and parietal cortices. We were also interested in whether lateralization of lesion (left versus right) would be preferentially related to lateralization of brain structures associated with outcome, given predominant lateralization of language within the left hemisphere [52].

## 2. Methods

### 2.1 Patient population

We identified 62 consecutive patients (35 men, mean age = 49±17.5) who underwent planned awake resections of primary (n = 55, 88%) and metastatic (n = 4, 6%) brain tumors, and symptomatic cavernous malformations, a benign tumor of blood vessels (n = 4, 6%) at the University of New Mexico Hospital between 2015 and 2019. Forty two of these patients had pre- and post-surgery neuropsychological testing, as well as baseline structural neuroimaging, allowing for comparison of brain structure to cognitive outcome in this subsample of our larger cohort. This study was approved by our institutional review board of the University of New Mexico Health Sciences Center (IRB) under study protocol #18–044. This study utilized chart review data obtained through routine clinical care, and waiver was obtained from the IRB for use of data for analysis.

### 2.2 Pre-and post-operative neuropsychological evaluation

All patients underwent a standard battery of neuropsychological tests, pre-operatively, to assess baseline cognitive functioning and to determine their ability to undergo awake resection of their tumor. Patients were administered a broad battery of tests including: Test of Premorbid Functioning (TOPF); Wechsler Abbreviated Scale of Intelligence 2nd Edition (WASI-II); Neuropsychological Assessment Battery Screening (NAB-Screening); Neuropsychological Assessment Battery (NAB-Language, Spatial, Executive); Processing Speed Index–Wechsler Adult Intelligence Scale–IV (PSI-WAIS-IV); Rey Complex Figure Test (RCFT); Brief Visual Memory Test Revised (BVMT-R); California Verbal Learning Test—II (CVLT-II); Controlled Oral Word Association Test (COWAT); Trail Making Tests–Parts A & B (TMT-A & B); Grip Strength (GRIP); Grooved Pegboard Test (PEG); Clinical Assessment of Depression (CAD); Wisconsin Card Sort Test (WCST); Trauma Symptom Inventory—II (TSI-II).

Post-operative testing was performed 4–6 months following surgery to ensure uniformity of data and allowing for time to recover from radiation-induced declines.

### 2.3 Imaging

All patients underwent a pre-operative standard clinical scan which included fMRI, FLAIR, T2, and T1 sequences. For our analysis we utilized the T1 sequences collected on a 3 T Phillips Ingenia scanner with the following sequence parameters: echo time TE = 3.5 ms, repetition time TR = 7.9 ms, flip angle = 8˚, voxel resolution = 0.9375 mm in plane, slice thickness = 1 mm. Three patients were scanned on a Siemens Symphony 1.5 T MRI scanner: TE = 4.68 ms, TR = 11 ms, flip angle = 20˚, voxel resolution = 1 mm in plane, slice thickness = 1 mm. Cortical reconstruction and volumetric segmentation were performed with the FreeSurfer-v6.0 image analysis suite, which included Talairach transformation, segmentation of subcortical and cortical structures, and production of values for cortical thickness, surface area, and volume [53–55]. Volume measures include a combination of thickness (a one-dimensional measure) and area (a two dimensional measure) across 33 measures per hemisphere (i.e., 66 across the

surface of the brain). All processed images were inspected for image quality. All tumor volumes were manually identified and masked by a neurosurgeon (M.C.) and re-processed to ensure accurate volumes. Tumor volume was highly skewed, with most being of low volume and a few patients having large volume tumors (Mean = 46,243 mm3, s.d. = 47,197 mm3). Twenty-five lesions were in the left hemisphere, while seventeen were in the right hemisphere. Finally, twenty lesions were within frontal lobe regions (including motor cortex), fourteen were within temporal lobe regions, six were within parietal cortex (including sensory cortex), and one was in "other" cortex (e.g., occipital).

## 2.4 Statistical analysis

Z-scores were created across major neuropsychological domains including: Total z-score, comprised the z-score inclusive of Attention, Memory, Language, Visuo-spatial, and Executive functioning. Mean and standard deviation scores were calculated across all subjects to derive z-score transformations [(Individual Score–Mean Score) / Standard Deviation = z-score)]. Particular subtests for each domain are as follows: *Attention*–Trail Making Test Part A, NAB Digit Span Forward, NAB Visual Attention Part B Efficiency; *Memory*–BVMT–Delayed Recall, CVLT–Delayed Recall; RCFT–Delayed Recall; *Language*–Boston Naming Test, COWAT Animal Naming; *Visuo-Spatial*–RCFT Copy Trial, Block Design (WASI), NAB Visual Designs; *Executive*–Trail Making Test Part B, COWAT (FAS), NAB Digits Backward.

Linear regression models were used to regress all cortical volumes from both hemispheres, against the Total z-score to determine the relationship between brain structure and broad cognitive functioning in the entire sample. We controlled for age, sex, and total segmented brain volume in each analysis. Family-wise p was held to $p < .05$ to control for Type I error given multiple comparisons [56]. We also separated the group by tumor lateralization (Left versus Right) to determine the linear relationship between brain structure and broad cognitive functioning in this subgroup of patient. Tumor volumes were measured by the neurosurgeon (MC), who was blind to performance on cognitive and mood tasks.

## 3. Results

We first evaluated the outcome of our patient group in terms of overall neuropsychological functioning, following tumor resection. For cognitive outcome, we used the difference score between total neuropsychological functioning at baseline (Total-Z-baseline) and total neuropsychological functioning at post-operative neuropsychological outcome (Total-Z-outcome). Our cohort consisted of 23 males and 19 females, of whom 25 patients had left hemisphere lesions, and 17 had right hemisphere lesions. There was no significant difference between males and females in terms of lesion lateralization (F = .182, p = .67). Overall, total neuropsychological functioning was somewhat worse at post-operative neuropsychological outcome (Mean = -.20) than at baseline (Mean = .00), the decrement of which was significant (t = 2.75, p = .009). Those patients who had radiation following resection (N = 32) performed marginally worse at neuropsychological outcome than those who did not receive radiation (N = 10) (F = 4.69, p = .036). Looking at subscale measures, patients experienced significant decrements across domains including attention (Mean = -.24; t = 3.03, p = .004), language (Mean = -.36; t = 3.09, p = .004), and executive functioning (Mean = -.26; t = 2.98, p = .005), while memory (Mean = -.02) and visuospatial (Mean = -.14) functioning were not significantly different between baseline and outcome measures.

We next explored the relationship between the difference score at baseline and outcome neuropsychological measures (Total-z-difference) and cortical volumes throughout the brain, controlling for age, sex, and total intracranial volume. We found that a model that included

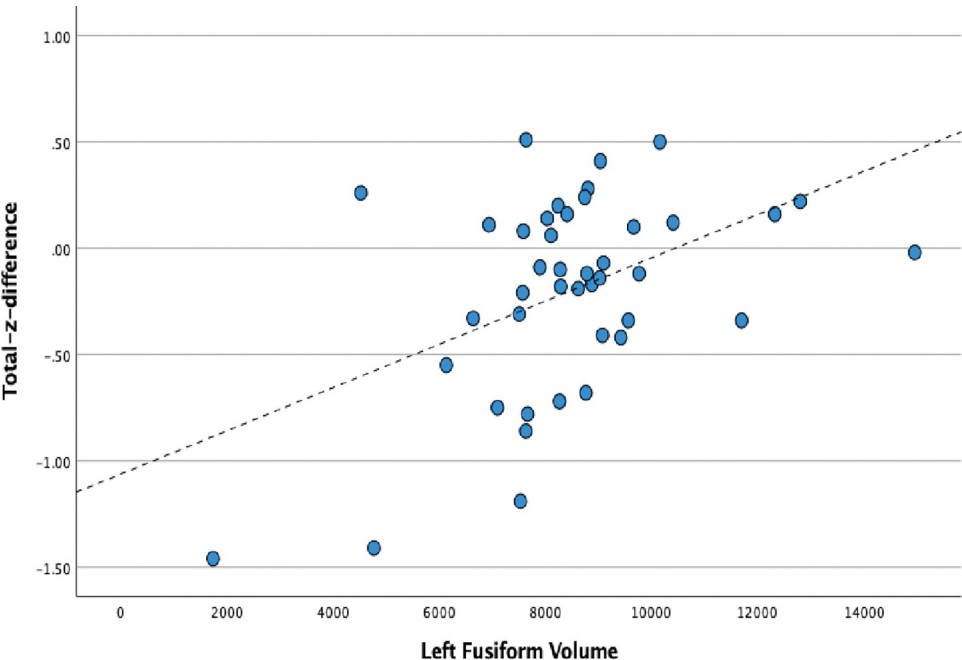

**Fig 1.**

volumes of the left fusiform, right lateral orbital frontal, right post central, and right paracentral volume predicted neuropsychological differences between baseline and outcome (F = 15.65, p = .001; r2 = .76). The bivariate relationship between left fusiform volume and Total-z-difference is presented in Fig 1.

Finally, we were interested whether lateralization of lesion was related to neuropsychological-volume relationships. For left lesion patients (N = 25), we found that a model including left fusiform, right pars orbitalis, right lateral fusiform volume and right cuneus, predicted Total-z-difference scores (F = 34.08, p < .001). The bivariate relationship between left fusiform volume and Total-z-difference score is presented in Fig 2.

In contrast, for right lesion patients, a model including only the right lateral orbital frontal volume predicted Total-z-difference (F = 7.76, p = .002) (Fig 3). Images of the left fusiform and right lateral orbital frontal regions are presented in Figs 2 and 3.

## 4. Discussion

Our study investigated the relationship between neuropsychological difference scores (pre- versus post-surgery), in patients undergoing awake craniotomy for tumor resection, and pre-surgical cortical tissue volumes. We found that several discrete brain volumes obtained pre-surgery predicted neuropsychological outcome post-resection. For the total sample, these volumes included: left fusiform, right lateral orbital frontal, right post central, and right paracentral regions. Our finding would imply either reserve capacity in such brain regions through neuroplastic cortical reorganization following removal of tumor and surrounding tissue [57], or could imply reorganization of large scale brain networks following network degradation/ reorganization via tumor growth and subsequent resection [58]. There is evidence for alteration [59] and reorganization [60] of cortical networks with glioma, and a better understanding of distinct cortical regions that could represent a functional "reserve capacity" is critical to preserve cognitive outcome following tumor resection. If brain function is dependent on

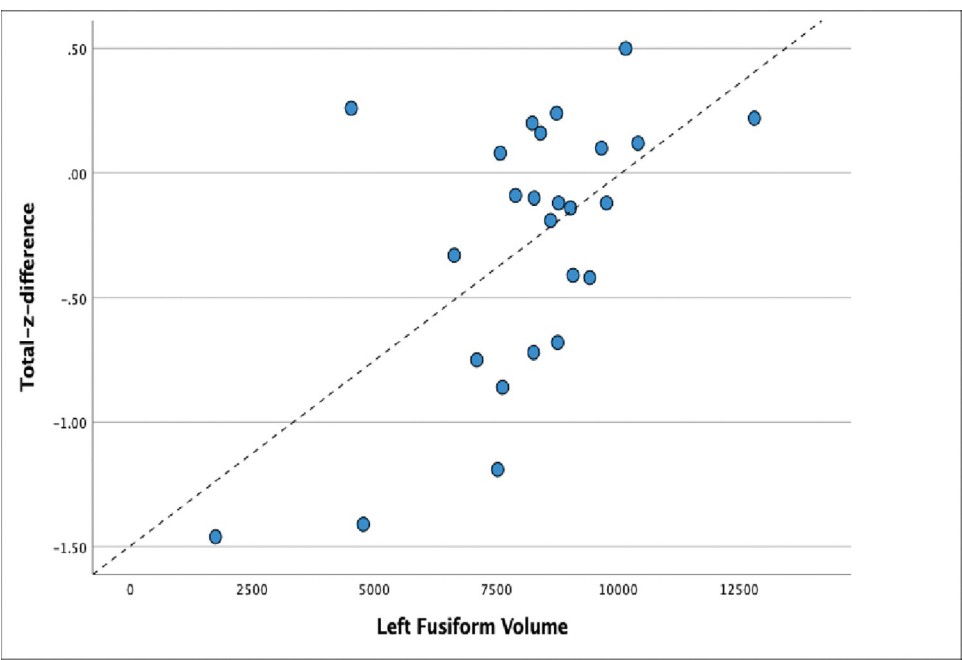

**Fig 2.**

network organization, and distinct cortical regions represent nodes within a network, then disruption of a cortical region (regardless of its "eloquence") is likely to degrade a wider range of brain connectivity [61].

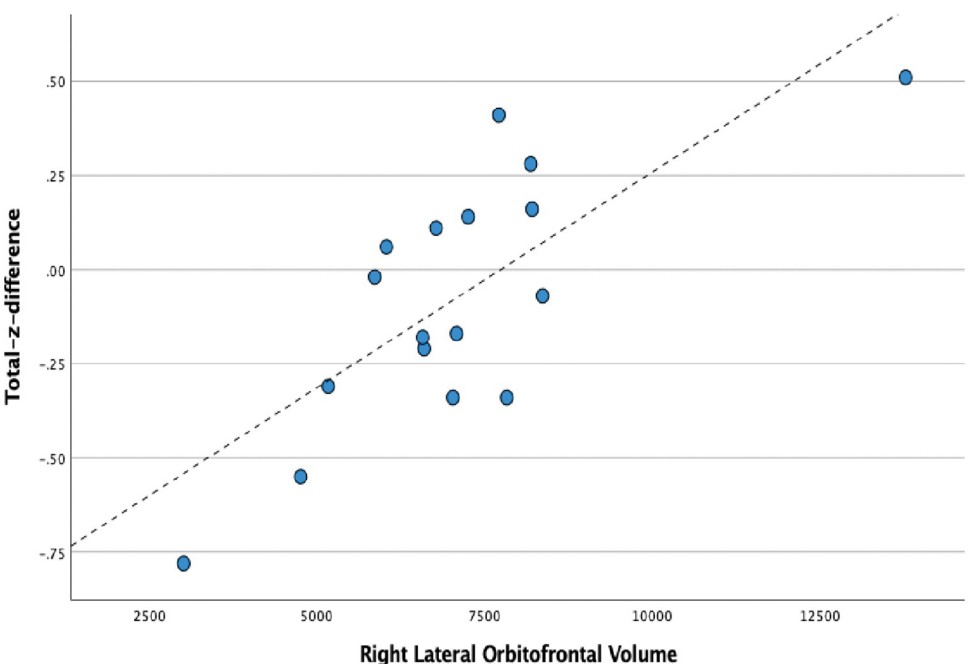

**Fig 3.**

Our main finding is that, regardless of lesion lateralization, volumes within the right frontal lobe, and specifically right orbitofrontal cortex, predicted neuropsychological difference scores (e.g., right lesion—right lateral orbital frontal; left lesion—right pars orbitalis), highlighting the importance of the right lateral orbital frontal cortex to neuropsychological outcome in this patient group. The lateral orbital frontal cortex (which includes BA 10, 11 and pars orbitalis or BA 47/12) [62] has been implicated in "stimulus-stimulus association learning", reward and punishment related behavior [63, 64], and "suppression of previously rewarded responses," [65]. In a large meta-analysis of connectivity modeling of the medial and lateral orbitofrontal cortex [66], the lateral orbital frontal cortex showed predominant co-activations with a network of brain regions within the prefrontal cortex, and other regions involved in language (e.g., pars orbitalis/ triangularis/opercularis) and memory functioning (e.g., hippocampus, medial thalamus, dorsal head of caudate). We hypothesize that the volume of tissue within the right lateral orbital frontal lobe (classically considered to be "non-eloquent" by neurosurgery standards) represents important reserve capacity in patients undergoing tumor surgery, representing a major connectivity hub subserving higher order cognitive functions subserving motivational learning, reward and punishment associations," semantics (e.g, language comprehension), and memory processing [62, 64, 65, 67, 68]. In left lateralized tumor patients, the left fusiform region was also related to neuropsychological difference scores. This region is most commonly associated with identification of visual objects, particularly faces [69] although research has associated this region with a multiplicity of functions including shape processing, object recognition, and reading [70].

There is controversy in the literature regarding the definition of EOR and how that impacts overall survival in brain tumor patients [71]. For example, in high grade gliomas, some groups have advocated removal of tissue beyond areas of contrast enhancement, i.e. the peritumoral zone represented by T2- fluid attenuated inversion recovery (FLAIR) regions on MRI [72–74], a concept called "supratotal resection". This FLAIR region is thought to represent the tumor infiltrative zone which drives future recurrence [75–77]. This "anatomical" supratotal resection is contrasted with a "functional" supratotal resection which advocates for pushing the resection until so-called "eloquent" tissue is encountered at both cortical and subcortical levels [78]. For example, in diffuse low grade gliomas, resections are carried beyond the FLAIR regions into normal tissue, so that "there is no margin left around these functional boundaries" [79, 80]. When done with awake mapping, such "functional" supratotal resections provide reasonable functional outcomes in both low grade [81, 82] and high grade gliomas [83]. We believe that cognitive deficits reflective of complex behavior are largely underappreciated in tumor patients.

A similar controversy exists in what exactly defines "eloquence" in the brain. Within the neurosurgical literature, functional areas have included sensorimotor, perisylvian language areas in the dominant hemisphere, visual cortex and deep subcortical structures including basal ganglia, internal capsule and thalamus [84]. Classically, areas not included in this definition are often considered "non-eloquent" [79], although this definition is evolving as increased structure-function data is gleaned through modern neuroimaging techniques. These include anterior frontal lobes including frontal pole, dorsolateral prefrontal cortex (pre-SMA), non-dominant temporal lobe, bilateral temporal poles and the non-dominant parietal cortex.

A gradual evolution is taking place within the neurosurgical community with an aim to shift the understanding of brain function (and hence, eloquence) from a "localizationist" view to a "network" or "connectomics" view, informed by modern neuroimaging findings highlighting complex structure function relationships throughout the brain. This paradigm shift is largely due to enormous advances made by the neuroscience community, for example the human connectome project [85, 86], but also realization on a wider scale that post-

operative deficits in higher cognitive functions (i.e. beyond hemiparesis and aphasia) are underappreciated in this patient population [87, 88]. This suggests that our current understanding of brain eloquence is insufficient, and that neurosurgeons are vulnerable to sacrificing brain tissue of vital importance to certain functional hubs within a network. We believe that our study provides an argument that reinforces this evolution of thinking.

There are several limitations of this study. Given the retrospective nature of the study, findings would have to confirmed in a larger prospective cohort. Imaging analysis only included baseline data, as post-operative imaging was obtained on different scanners, thus precluding uniformity of analysis. Moreover, the addition of adjuvant chemotherapy and radiation treatment is likely to have had effects on cognitive functioning as well as brain structure [89, 90]. Our main focus was total neuropsychological outcome; limitations in population size did not allow for correlations of individual cognitive domains with brain structure. More long-term data (beyond a year) would be needed to see whether associations found in our study continue to predict neuropsychological outcome in these patients.

## Supporting information

**S1 File. Raw data file: AwakePre-post.**
(XLSX)

## Author Contributions

**Conceptualization:** Muhammad Omar Chohan, Rex Eugene Jung.

**Data curation:** Muhammad Omar Chohan, Ranee Ann Flores, Christopher Wertz, Rex Eugene Jung.

**Formal analysis:** Muhammad Omar Chohan, Ranee Ann Flores, Christopher Wertz, Rex Eugene Jung.

**Investigation:** Muhammad Omar Chohan, Christopher Wertz, Rex Eugene Jung.

**Methodology:** Muhammad Omar Chohan, Christopher Wertz, Rex Eugene Jung.

**Project administration:** Muhammad Omar Chohan, Ranee Ann Flores, Rex Eugene Jung.

**Resources:** Muhammad Omar Chohan.

**Supervision:** Rex Eugene Jung.

**Validation:** Ranee Ann Flores.

**Writing – original draft:** Muhammad Omar Chohan, Ranee Ann Flores, Christopher Wertz, Rex Eugene Jung.

**Writing – review & editing:** Muhammad Omar Chohan, Rex Eugene Jung.

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
