## [Decision Letter · Decision Letter 0]

27 Jan 2023

PONE-D-22-27442“Non-Eloquent” brain regions predict neuropsychological outcome in tumor patients

undergoing awake craniotomyPLOS ONE

Dear Dr. Jung,

Thank you for submitting your manuscript to PLOS ONE. After careful consideration, we feel that it has merit but does not fully meet PLOS ONE’s publication criteria as it currently stands. Therefore, we invite you to submit a revised version of the manuscript that addresses the points raised during the review process.

We look forward to receiving your revised manuscript.

Kind regards,

Irene Cristofori

Academic Editor

PLOS ONE

Journal Requirements:

3. Please amend your manuscript to include your abstract after the title page.

Reviewers' comments:

Reviewer's Responses to Questions

**Comments to the Author**

1. Is the manuscript technically sound, and do the data support the conclusions?

Reviewer #1: Yes

2. Has the statistical analysis been performed appropriately and rigorously? 

Reviewer #1: Yes

3. Have the authors made all data underlying the findings in their manuscript fully available?

Reviewer #1: No

4. Is the manuscript presented in an intelligible fashion and written in standard English?

Reviewer #1: Yes

5. Review Comments to the Author

Reviewer #1: The authors present a retrospective study of 62 patients who underwent awake neurosurgery for brain tumors. They report the effect of brain lesions on the neuropsychological evolution of patients. They conclude that lesions affecting so-called non-eloquent brain regions can affect some high order human functions.

I enjoy reading the manuscript as it is well written and organised.

I would like the authors to clarify several points before the manuscript can be published.

Major:

1) Overall, the manuscript just seems to confirm facts that are all ready well known by the scientific community. Maybe less by neurosurgeons but very well known by some and by neuroscientists. Indeed, the concept of eloquent brain regions as motor, sensory, speech regions is out of date. High order human functions are far more complex and organised in a wide interconnected brain network. For example, the role of the prefrontal/frontal cortex in general has been extensively explored, and even though it is not commonly refered as an "eloquent area" from an "classical" neurosurgical perspective, it known to be a critical brain region for many functions. Therefore, the authors should temper a little the overall novelty of their findings.

2) did the authors use correction for multiple analysis ? Indeed as they are many many brain regions (so many many tests) so the statistical value of significance should be corrected for multiple analysis. Then the authors should confirm or not that the brain region they stated significant "stay" significant with this stastistical methods.

minor

1) the authors should give more details on the tumor included (volume, exact localisation (brain region involded)...)

2) why awake was deemed necessary for these patients in the first place ?

3) what was the peroperative testing protocol ? Were high order functions tested during the awake phase (especially since some functions were altered pre operatively)?

6. PLOS authors have the option to publish the peer review history of their article (what does this mean?). If published, this will include your full peer review and any attached files.

Reviewer #1: No

---

## [Author Response · Author response to Decision Letter 0]

14 Mar 2023

Reviewer #1: The authors present a retrospective study of 62 patients who underwent awake neurosurgery for brain tumors. They report the effect of brain lesions on the neuropsychological evolution of patients. They conclude that lesions affecting so-called non-eloquent brain regions can affect some high order human functions.

I enjoy reading the manuscript as it is well written and organised.

We thank the reviewer for this positive comment.

I would like the authors to clarify several points before the manuscript can be published.

Major:

1) Overall, the manuscript just seems to confirm facts that are all ready well known by the scientific community. Maybe less by neurosurgeons but very well known by some and by neuroscientists. Indeed, the concept of eloquent brain regions as motor, sensory, speech regions is out of date. High order human functions are far more complex and organised in a wide interconnected brain network. For example, the role of the prefrontal/frontal cortex in general has been extensively explored, and even though it is not commonly refered as an "eloquent area" from an "classical" neurosurgical perspective, it known to be a critical brain region for many functions. Therefore, the authors should temper a little the overall novelty of their findings.

We agree that the term is *certainly* dated given the knowledge provided by modern neuroscience and neuropsychology. We have attempted to temper the novelty in the manuscript where appropriate, particularly by noting that this term is classically considered to be “non-eloquent” by neurosurgeons, with more modern neuropsychological, neuroimaging, and neuropsychiatric data implicating these regions in numerous higher cognitive processes.

2) did the authors use correction for multiple analysis ? Indeed as they are many many brain regions (so many many tests) so the statistical value of significance should be corrected for multiple analysis. Then the authors should confirm or not that the brain region they stated significant "stay" significant with this stastistical methods.

We thank the reviewer for this important comment. We have utilized a family-wise correction for this exploratory paper (we state in the manuscript: “Family-wise p was held to p <.05 to control for Type I error given multiple comparisons”). That is, we adopt a .05 statistical significance threshold under which all major contrasts must fall below (in total) – a family wise error rate (FWER) using ordered p values (the Holm procedure – Holm 1979) Holm, S. (1979). A simple sequentially rejective multiple test procedure. Scandinavian Journal of Statistics, 6, 65-70. Thus, the overall confidence of all contrasts being true is held at 95%. We believe that this, more lenient, correction is appropriate for this exploratory study as compared to the stricter Bonferroni, which allocates the false discovery equally (Bonferroni) across contrasts as opposed to sequentially (Holm)(one contrast can be lenient – i.e., .04, but the rest must be increasingly stricter – i.e., .01 each, until the .05 threshold is reached). We have included the Holm reference to make note of our approach.

Minor

1) the authors should give more details on the tumor included (volume, exact localisation (brain region involded)...)

Tumor volume was highly skewed, with most being of low volume and a few patients having large volume tumors (Mean = 46,243 mm3, s.d. = 47,197 mm3). Twenty-five lesions were in the left hemisphere, while seventeen were in the right hemisphere. Finally, twenty lesions were within frontal lobe regions (including motor cortex), fourteen were within temporal lobe regions, six were within parietal cortex (including sensory cortex), and one was in “other” cortex (e.g., occipital). We have added this text to the manuscript in the Imaging section.

We have added a figure showing the tumor volumes by patient for the reviewer’s information. We do not feel that this figure adds significantly to the manuscript, but are happy to add it if the reviewer feels otherwise.

2) why awake was deemed necessary for these patients in the first place ?

Awake surgery was determined to be clinically necessary if behavioral function was determined to be likely to decline significantly due to tumor resection based on 1) location of tumor to so-called “eloquent” tissue, 2) neuropsychological status, and 3) neurosurgical input. Such tissue was often localized to sensory, motor, and language networks in both left and right hemispheres based on fMRI and neuropsychological testing.

3) what was the peroperative testing protocol ? Were high order functions tested during the awake phase (especially since some functions were altered pre operatively)?

Primarily language and motor functioning was testing during awake procedure, with stimulation and resection extension progressing to language and motor/sensory networks. Occasional testing was done for other domains (e.g., attention, executive, visual, etc.), although this was not done in as systematic fashion as was language and motor testing across the coho

---

## [Editor Report · Decision Letter 1]

28 Mar 2023

“Non-Eloquent” brain regions predict neuropsychological outcome in tumor patients undergoing awake craniotomy

PONE-D-22-27442R1

Dear Dr. Jung,

We’re pleased to inform you that your manuscript has been judged scientifically suitable for publication and will be formally accepted for publication once it meets all outstanding technical requirements.

Kind regards,

Irene Cristofori

Academic Editor

PLOS ONE
---

## [Editor Report · Acceptance letter]

4 Apr 2023

PONE-D-22-27442R1 

“Non-Eloquent” brain regions predict neuropsychological outcome in tumor patients undergoing awake craniotomy 

Dear Dr. Jung:

I'm pleased to inform you that your manuscript has been deemed suitable for publication in PLOS ONE. Congratulations! Your manuscript is now with our production department. 

Kind regards, 

on behalf of

Dr. Irene Cristofori 

Academic Editor

PLOS ONE